# An Enhanced Feature Pyramid Object Detection Network for Autonomous Driving

**Yutian Wu [1],\*** , **Shuming Tang [2]**, **Shuwei Zhang [1] and Harutoshi Ogai [1]**

[1]  Graduate School of Information, Production and Systems, Waseda University, Fukuoka 8080135, Japan; ZHANGSHUWEI@toki.waseda.jp (S.Z.); ogai@waseda.jp (H.O.)

[2]  Institute of Automation, Chinese Academy of Sciences, Beijing 100190, China; shuming.tang@ia.ac.cn

\*  Correspondence: wuyutian@fuji.waseda.jp

**Abstract:** Feature Pyramid Network (FPN) builds a high-level semantic feature pyramid and detects objects of different scales in corresponding pyramid levels. Usually, features within the same pyramid levels have the same weight for subsequent object detection, which ignores the feature requirements of different scale objects. As we know, for most detection networks, it is hard to detect small objects and occluded objects because there is little information to exploit. To solve the above problems, we propose an Enhanced Feature Pyramid Object Detection Network (EFPN), which innovatively constructs an enhanced feature extraction subnet and adaptive parallel detection subnet. Enhanced feature extraction subnet introduces Feature Weight Module (FWM) to enhance pyramid features by weighting the fusion feature map. Adaptive parallel detection subnet introduces Adaptive Context Expansion (ACE) and Parallel Detection Branch (PDB). ACE aims to generate the features of adaptively enlarged object context region and original region. PDB predicts classification and regression results separately with the two features. Experiments showed that EFPN outperforms FPN in detection accuracy on Pascal VOC and KITTI datasets. Furthermore, the performance of EFPN meets the real-time requirements of autonomous driving systems.

**Keywords:** object detection; feature pyramid network; feature recalibration; context embedding; autonomous driving systems; augmented reality

---

## 1. Introduction

Autonomous driving systems cognize the surrounding environment by modeling the street scenery information acquired by sensors, and then makes driving decisions. Among them, object detection plays an important role in the construction of realistic scenes. In recent years, the improvement of deep learning theory and computing power has significantly accelerated the development of object detection. Nowadays, object detection methods are divided into two branches, one-stage and two-stage. Two-stage algorithms are widely applied in tasks with a high degree of accuracy. A typical two-stage deep learning-based detection framework includes a feature extraction subnet, a proposal extraction subnet, and a detection subnet. The feature extraction subnet [1–3] uses a convolutional neural network to extract features with high robustness and rich information through end-to-end training process. The proposal extraction subnet [4,5] generates regions of interest (RoIs), including foreground positive samples and background negative samples from the feature map. The detection subnet utilizes the pooling feature of RoIs to predict the classification and regression results of detected objects [5–9].

To utilize the features obtained from the feature extraction subnet, initially, researchers feed the single scale features from the last convolution layer into the subsequent subnet [5,7,8]. Although these features are rich in higher-level semantic information, they lack detailed information. This leads to a

poor detection performance for small objects and occluded objects. Two kinds of network structures are proposed to solve this problem. One is to do prediction separately on feature layers of different resolutions [10,11]. The other is to merge multi-resolution features at first and then to predict on the merged feature map [12,13]. Further studies show that taking advantages of both structures can get more accurate results [14–17]. FPN is a state-of-the-art network of such structure. FPN constructs a feature pyramid with high-level semantics throughout and independently predicts at each pyramid level. The key design for building feature maps for each pyramid level is the lateral connection. It merges the semantically stronger feature maps from top-down with feature maps which are rich in detail localization information from the same bottom-up level. However, feature maps within the same pyramid level have the same weight for subsequent object detection, which ignores the feature requirements of different scale objects. Inspired by the features strengthen block in image classification [18] and segmentation [19] field, we propose a generic FWM to recalibrate feature maps in each pyramid level. For each pyramid layer, learnable weights are applied between features along space and channel to boost useful features and suppress useless features specific for different scale detection requirements. After that, a new enhanced feature pyramid is reconstructed with more powerful feature representation.

After calculating the shared features, the proposal extraction subnet generates RoIs. Then, the detection subnet adopts RoI pooling to map and extract the corresponding fixed-size features of RoIs for classification and regression. However, for small and occluded objects whose feature information is limited by their size or physical occlusion, their RoI features are less informative for subsequent prediction. Researchers argue that using image evidence beyond RoI content benefits object detection [10,13,20–23] and segmentation [21]. Such context information can be integrated by a spatial recurrent neural network or acquired by expanding or deforming original RoI regions. We mainly focus on local context embedding methods by expanding RoI regions. Gidaris et al. [20] defined ten kinds of context regions around RoI to enrich the RoI representation, and then integrated each region's information in separate fully connected layers and concatenated them together as final features of each RoI. It does improve accuracy at the cost of calculating a lot of redundant and overlapping context blocks, which is incompatible with real-time autonomous driving systems. Cai et al. [10] stacked features of a RoI and its larger context RoI together, and then compressed them by a convolution operation as a merged RoI representation. However, all the RoIs share a fixed context expansion ratio, which is set by human and requires careful tuning. This causes insufficient or redundant context information supplementation of various scale objects. Wang et al. [22] introduced local competition mechanism to select the most useful context region among three different expansion ratios of RoIs, but all of the expansion values are still set manually. Since each object has different requirements for context information, the amount of context complement should be distinct between each RoI. For instance, it is difficult to detect small objects without referring to the surrounding context, while the pure object characteristics is what a detection model should focus on in the case of large objects. To meet this requirement, we construct an adaptive parallel detection subnet, which innovatively introduces Adaptive Context Expansion(ACE) and Parallel Detection Branch (PDB). We naturally leverage the pyramid shape of FPN to adaptively supply the context information of RoI of different sizes. For example, we provide more context information for low-resolution small object while less for large object. Then, features of context expansion RoI (ceRoI) and original RoI, respectively, serve as the input of PDB to separately predict classification and regression results. With the parallel design, we can make the best of two features, which not only maintains the location accuracy but also improves classification performance.

The rest of this paper is organized as follows. In Section 2, we introduce our proposed EFPN with an innovative enhanced feature extraction subnet and an adaptive parallel detection subnet. In Section 3, we evaluate and discuss the object detection results of EFPN on several open datasets. In Section 4, we introduce the application of EFPN as vision-based object detection module in an autonomous driving cart system. Finally, conclusions are drawn in Section 5.

## 2. Proposed Method

EFPN is our proposed object detection network. Its architecture is shown in Figure 1. Firstly, in enhanced feature extraction subnet, we generate pyramid features in the same way as FPN. Features in each pyramid level are weighted by our proposed FWM, and a new enhanced feature pyramid is reconstructed as the input for the following procedure. Secondly, in the proposal extraction subnet, Region Proposal Network (RPN) [5] is used to generate anchors of various shapes on the enhanced pyramidal feature map. Thirdly, in adaptive parallel detection subnet, ACE is applied to extract the feature of ceRoI and RoI for each foreground RoI. Two kinds of RoI features are, respectively, fed into PDB to predict classification and regression as the final detection results.

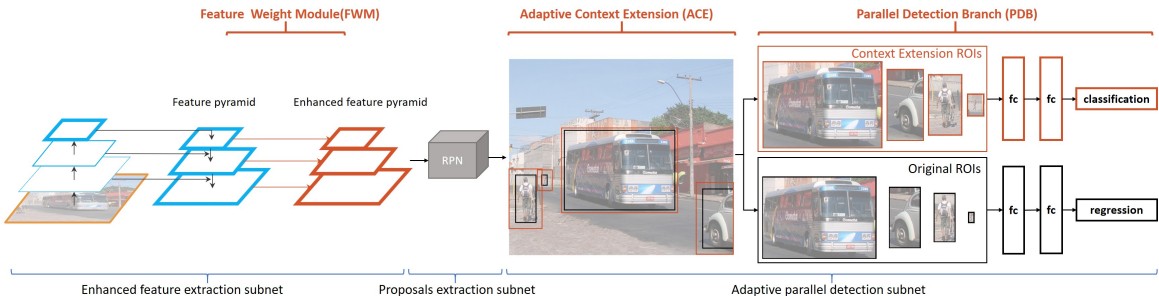

**Figure 1.** The structure of EFPN.

### 2.1. Enhanced Feature Extraction Subnet

Generally, FPN first builds the bottom-up layers $\{C_2, C_3, C_4, C_5\}$ by the feedforward computation of backbone ConvNet. Then, FPN constructs each top-down feature maps by element-wised adding the top-down feature maps of the last pyramid level with the bottom-up feature maps of the same pyramid level, which is shown in t Figure 2 (left). The set of pyramidal feature maps built by FPN is $\{P_2, P_3, P_4, P_5\}$.

Despite such a careful design for generating refined merged feature maps for different levels, it is not strong enough for the information of spatial and channel features to different scaled objects. We hypothesize that both spatial-wise and channel-wise recalibrating merged feature maps can encourage current pyramid layer detection. Hence, we propose FWM to enhance the pyramid feature. The structure of FWM is shown in Figure 2 (right).

FWM starts by modeling the feature dependency of the feature maps in each pyramid level, and further learns the feature importance vector to recalibrate the feature maps to emphasize the useful features. Specially, FWM in each pyramid level is in the same structure but has different learnable weights, which results in different calculated feature weights. Each FWM consists of three sub-modules: Feature Channel Weight Module (FCWM), Feature Spatial Weight Module (FSWM) and Feature Channel Spatial Weight Module (FCSWM). FCWM and FSWM calculate the feature importance vector along channel and spatial location. FCSWM combines the recalibrated weighted feature maps after FCWM and FSWM as the new pyramidal feature maps. The detailed design of the three submodules are described in the following subsections.

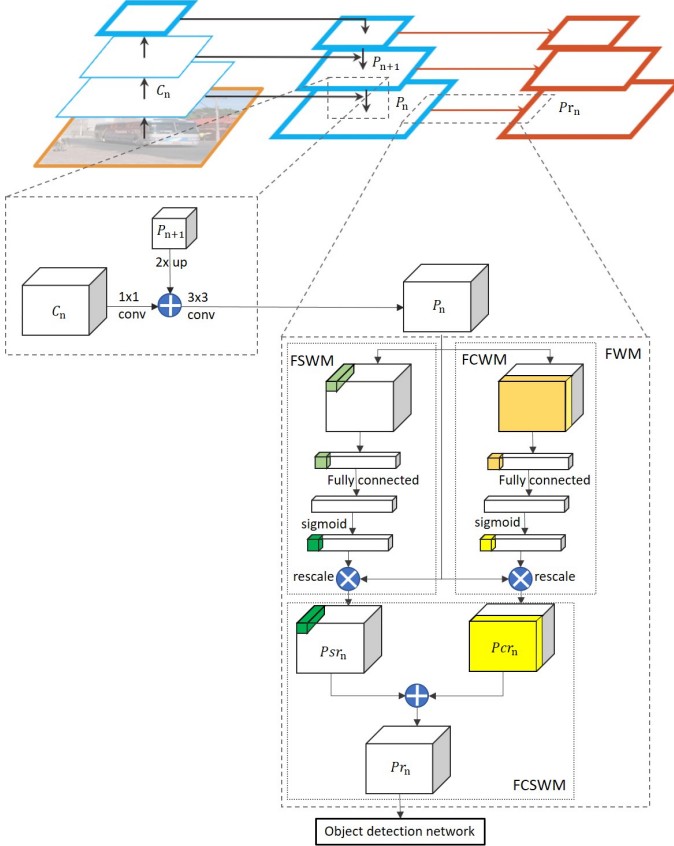

**Figure 2.** The structure of enhanced feature extraction subnet.

### 2.1.1. Feature Channel Weight Module (FCWM)

FCWM focuses on enhancing features along channel of each pyramid level. FCWM first explicitly models the dependency of features along channel and learns a channel specific descriptor through the squeeze-and-excitation method [18]. Then, it emphasizes the useful channels for more efficient global information expression of feature maps in each pyramid level.

Suppose the feature maps in $n$th pyramid level is $P_n$, which is generated by FPN. $H_n$ and $W_n$ are the spatial height and width of $P_n$, respectively. The $i$th channel feature is $P_n^i$.

At the beginning, we do global average pooling on $P_n^i$ to get the global distribution response $Z_n^i$:

$$Z_n^i = \frac{1}{H_n \times W_n} \sum_p^{H_n} \sum_q^{W_n} P_n^i(p, q). \tag{1}$$

We use two fully connected layers to map the non-linear correlation between all global distribution responses $Z_n$ and obtain the feature importance vectors $\hat{Z}_n$:

$$\hat{Z}_n = W_n^1(\delta(W_n^2 Z_n)), \tag{2}$$

where $W_n^1$ is the weight of the first fully connected layer. $W_n^2$ is the weight of the second fully connected layer. $\delta$ represents the ReLU function.

Then, we normalize $\hat{Z}_n$ to $[0, 1]$ as a weight vector:

$$r_n = \sigma(\hat{Z}_n), \tag{3}$$

where $\sigma$ represents Sigmoid function.

Finally, we assign the weight $r_n$ to the original feature $P_n$ and get the new pyramid feature $Pcr_n$ after channel-wised recalibration:

$$Pcr_n = P_n r_n = [P_n^1 r_n^1, P_n^2 r_n^2, ..., P_n^n r_n^n]. \tag{4}$$

### 2.1.2. Feature Spatial Weight Module (FSWM)

Similar to the design of FCWM, FSWM enhances the features along spatial location of each pyramid level, which emphasizes the effective pixels and depresses the ineffective or low-effect pixels.

We define $P_n^{(p,q)}$ as the clipping of all channel features at each feature point $(p,q)$ of $P_n$. First, we integrate all the features of each point through a convolution operation to get the spatial importance vector $O_n^{(p,q)}$:

$$O_n^{(p,q)} = W_n^3 P_n^{(p,q)}, \tag{5}$$

where $W_n^3$ is the convolution kernel weight.

Then, we normalize $\hat{O}_n$ to $[0,1]$ as a weight vector $t_n$

$$t_n = \sigma(\hat{O}_n), \tag{6}$$

where $\sigma$ represents Sigmoid function.

Finally, the normalized weights are spatially weighted to each pixel to get the new feature $Psr_n$:

$$Psr_n = P_n t_n = [P_n^{(1,1)} t_n^{(1,1)}, P_n^{(1,2)} t_n^{(1,2)}, ..., P_n^{(H_n, W_n)} t_n^{(H_n, W_n)}]. \tag{7}$$

### 2.1.3. Feature Channel Spatial Weight Module (FCSWM)

FCSWM combines the channel-wised weighted $Pcr_n$ obtained by FCWM and the spatially weighted $Psr_n$ obtained by FSWM to generate a new recalibrated feature $Pr_n$. The combination operation is implemented by addition:

$$Pr_n = Pcr_n + Psr_n. \tag{8}$$

$Pr_n$ encourages original feature maps to be both spatial-wise and channel-wise more informative. In EFPN, we replace the initial feature pyramid features $\{P_2, P_3, P_4, P_5\}$ by the recalibrated enhanced pyramid features $\{Pr_2, Pr_3, Pr_4, Pr_5\}$ as the input feature of proposal extraction subnet and detection subnet.

### 2.2. Adaptive Parallel Detection Subnet

To inject object context information, we design adaptive parallel detection subnet, as shown in Figure 1. Adaptive parallel detection subnet includes ACE and PDB. ACE calculates the context region extension ratio of each RoI, and then generates and extracts the feature of ceRoI and original RoI after RoI pooling. PDB inputs the feature of ceRoI and RoI separately to object classification branch and regression branch to predict the classification and regression as the final detection results.

### 2.2.1. Adaptive Context Expansion (ACE)

As we know, small objects need extra information to help detection. FPN assigns small RoIs to finer-resolution level to add extra detailed RoI feature. Suppose a RoI has width $w$ and height $h$ (on the input image of the network); the level $k$ of RoI is determined by its area size $S$:

$$S = wh, \tag{9}$$

$$k = \lfloor k_0 + \log_2(\sqrt{S}/S_0) \rfloor. \tag{10}$$

Similarly, in ACE, an extra context feature is embedded for adding surrounding information for small RoIs. We define two context expanding criteria, which guide the calculation of context region expansion ratio for each RoI.

*Vertical expansion criterion* leverages the hierarchical structure of pyramid network to calculate the vertical context expansion ratio $R_v$:

$$R_v = \alpha k, \tag{11}$$

where $\alpha$ denotes the vertical context enlarge coefficient. Since context expending of RoI is positively correlated with the level of RoI, $\alpha$ is always positive. In Figure 3, the small $RoI_b$ and the large $RoI_c$ belong to different pyramid levels, hence their $R_v$ are clearly distinct from each other, that is a larger one for $RoI_b$ and a smaller one for $RoI_c$.

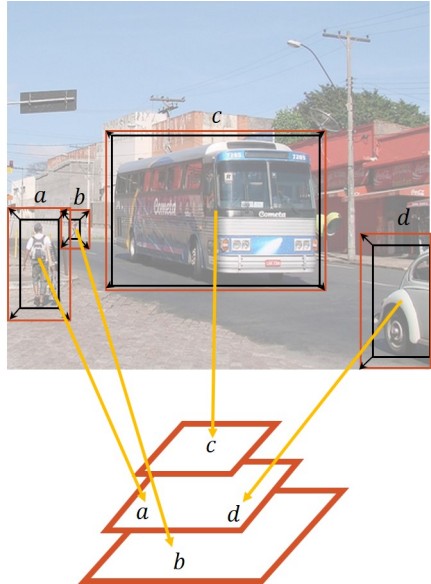

**Figure 3.** The structure of Adaptive Context Expansion (ACE).

*Horizontal expansion criterion* focuses on the calculation of refined expansion ratios between different objects on the same pyramid level. Although RoIs within the same scale range have the same $R_v$, their varying sizes lead to different requirements. We take this factor into account to provide more precise horizontal context expansion ratio $R_h$:

$$R_h = \beta(\log_2(\sqrt{S}/S_0) + k_0 - k), \tag{12}$$

where $\beta$ denotes the horizontal context enlarge coefficient. $\log_2(\sqrt{S}/S_0) + k_0 - k$ is the residual part of level vector, ranging $[0, 1]$, which indicates the size difference among the RoIs of the same pyramid level. In Figure 3, for $RoI_a$ and $RoI_d$ of the same pyramid level, the smaller its area is, the more context information it needs. Thus, they have slightly different $R_h$, with a larger one for $RoI_a$ and a smaller one for $RoI_d$.

Above all, the final context expansion ratio $R$, which considers the two independent criteria of each RoI, is:

$$R = R_v + R_h + \gamma, \tag{13}$$

where $\gamma$ is used to adjust magnitude of extension ratio among different dataset resulting from factors, such as background complexity, objects density, etc.

Next, the scale-up operation $\Theta$ based on the center point of each RoI is used to get the context region ceRoI:

$$ceRoI = \Theta(RoI), \tag{14}$$

where the ceRoI has the width of $w \times R$ and height of $h \times R$ with the same center as the original RoI.

### 2.2.2. Parallel Detection Branch (PDB)

ceRoI enriches the feature of RoI by aggregating its context environment information around, which will be helpful for object recognition. However, ceRoI changes the geometric position representation of original RoI. The misalignment of two features may somehow cause negative effects to sensitive bounding box regression.

To make better use of the context information, unlike most popular detection subnets, we design a divide-and-conquer prediction strategy, as shown in Figure 1. Features of ceRoI and RoI after RoI pooling are extracted and separately fed into two parallel branches. Each branch includes two hidden 1024D fully connected layers, which are attached by ReLU for object classification and regression.

### 2.2.3. Loss Function

When training the adaptive parallel detection subnet, the parameters $W$ are randomly initialized. Then, $W$ are learned from a set of training samples $S = \{(X_i, Y_i)\}_{i=1}^{N}$, where $X_i$ is a training RoI, and $Y_i = (y_i, b_i)$ is the combination of its class label $y_i \in 0, 1, 2, ...K$ and bounding box coordinates $b_i = (b_i^x, b_i^y, b_i^w, b_i^h)$. $K$ is the number of classes.

We use a multi-task loss $\mathcal{L}$ on each labeled $X_i$ to jointly train for classification and regression:

$$\mathcal{L}(X, Y | W) = \mathcal{L}_{cls}(p(\Theta(X)), y) + \lambda[y \geq 1]\mathcal{L}_{loc}(\hat{b}, b), \tag{15}$$

where $p(X) = (p_0(X), ..., p_K(X))$ is the probability distribution over classes, $\lambda$ is a trade-off coefficient, $\mathcal{L}_{cls}(p(\Theta(X)), y) = -\log p_y(\Theta(X))$ is the cross-entropy loss, $\hat{b} = (\hat{b}_x, \hat{b}_y, \hat{b}_w, \hat{b}_h)$ is the regressed bounding box, and

$$\mathcal{L}_{loc}(\hat{b}, b) = \frac{1}{4} \sum_{j \in \{x, y, w, h\}} smooth_{L_1}(\hat{b}_j, b_j) \tag{16}$$

is the smoothed bounding box regression loss [7]. The bounding box loss is only used for positive samples and the optimal parameters $W^* = argmin_W \mathcal{L}(W)$ are learned by stochastic gradient descent optimizer.

## 3. Experiments on Open Datasets

We evaluated our method using three challenging open datasets.

*Pascal VOC* [24] contains 20 categories of indoor and outdoor objects class. We mainly focused on the average precision (AP) of six classes appearing in road scenes and the mean average precision (mAP) of the whole dataset. We used VOC07+12 dataset, which contains 16551 images for training and 4952 images for testing to evaluate our proposed method. To further investigate the effectiveness of each module structure, we used VOC07 dataset, which contains 5011 training images and 4952 testing images for ablation study.

*KITTI* [25] is a large automatic driving dataset. Here, we used its 2D object detection dataset for evaluation. In our experiment, we redefined the object classes into two classes: *car* and *pedestrian*. *car* includes [*Van, Truck, Car, Tram*] and *pedestrian* includes [*Personsitting, Pedestrian, Cyclist*]. KITTI provides 7481 images for training and 7518 for testing. Since no ground truth is available for the test set, we split the training set into training set and validation set by 8:1.

*Cityscapes* [26] is a road scene image segmentation dataset. Pixel-level annotation for segmentation task contains more small and occluded labeled objects than detection datasets, which is full of challenges. Therefore, we converted Cityscapes into detection dataset to further test our model. The definition of object class is the same as that of KITTI. Cityscapes detection dataset consists of 2842 images.

### 3.1. Implementation Details

We implemented EFPN in Python within Pytorch deep-learning framework. Following Lin et al. [14], we resized each image so that its shorter side has 600 pixels. The network was trained using one NVIDIA RTX 2080 GPU with 1 image per mini-batch.

For proposal extraction subnet, we adopted the same design and training parameters as FPN.

For adaptive parallel detection subnet, we adopted RoI align [27] as RoI Pooling mechanism. We used a weight decay of 0.0001 and a momentum of 0.9. The learning rate started from 0.001 and was divided by 10 at every five epochs. The model had 10 total epochs.

For the parameter setting in ACE, we set $k_0$ to 4 and $S_0$ to 224 in Equation (10), consistent with the canonical ImageNet pre-training size. We set the $[\alpha, \beta, \gamma]$ in Equation (13) for different datasets: [0.1, 0.01, 0.6] for VOC07+12, [0.2, 0.01, 1.1] for VOC07, [0.15, 0.1, 0.9] for KITTI, and [0.15, 0.1, 0.9] for Cityscapes.

### 3.2. Object Detection Results on Pascal VOC

To evaluate the performance of EFPN and the two new subnets proposed in this paper, we compared them with other related object detection algorithms on VOC07+12 dataset. Results are shown in Table 1. Note that Approaches (b)–(d) and (i) were implemented and tested on the same platform as the proposed EFPN, while the results of Approaches (e)–(h) and (j) were from their corresponding publications.

**Table 1.** Object detection results evaluated on VOC07+12.

| Approach | Backbone | Car | Person | Bus | Bike | Motorbike | Train | mAP |
|---|---|---|---|---|---|---|---|---|
| (a) EFPN | Res101 | 88.7 | 85.4 | 88.4 | 86.8 | 88.2 | 88.0 | 81.6 |
| (b) FPN with enhanced feature extraction subnet | Res101 | 88.6 | 85.4 | 86.7 | 86.6 | 89.0 | 86.4 | 81.3 |
| (c) FPN with adaptive parallel detection subnet | Res101 | 88.5 | 84.4 | 88.0 | 88.5 | 86.4 | 86.9 | 81.4 |
| (d) FPN | Res101 | 88.2 | 84.7 | 86.9 | 85.5 | 85.5 | 87.2 | 81.1 |
| (e) SSD 513 | Res101 | 88.1 | 83.0 | 88.2 | 87.6 | 87.5 | 87.2 | 80.6 |
| (f) DSSD 513 | Res101 | 88.7 | 83.7 | 89.0 | 86.2 | 87.5 | 85.7 | 81.5 |
| (g) R-FCN | Res101 | 88.5 | 81.2 | 86.8 | 87.2 | 79.9 | 85.9 | 80.5 |
| (h) MR-CNN | VGG | 85.9 | 76.4 | 88.0 | 84.1 | 85.0 | 85.0 | 78.2 |
| (i) Faster R-CNN | Res101 | 85.3 | 75.4 | 85.1 | 80.7 | 80.9 | 85.3 | 76.4 |
| (j) ION | VGG | 85.1 | 74.4 | 85.4 | 83.1 | 82.2 | 84.2 | 75.6 |

Compared with baseline (d), the controlled experiments (b) and (c), which separately replace one part of the original FPN model, prove the validity of the proposed enhanced feature extraction subnet and adaptive parallel detection subnet. Since the two subnets enhance the feature by applying weight distribution of the entire feature map and feature supplementary of each RoI, respectively, they do not overlap or inhibit each other. This is evidenced by the increased accuracy of the merged method (a) compared with either single module (b) or (c). Overall, among all related methods, the proposed EFPN (a) holds the highest mAP and highest AP in car, person and train.

Adaptive parallel detection subnet assembles two new module ACE and PDB. We designed ablation studies to quantify the effect of the two modules on the VOC07 dataset. The results are reported in Table 2.

**Table 2.** Ablation study of the proposed ACE and PDB in adaptive parallel detection subnet on VOC07.

| Approach | Context Expansion | Detection Branch | mAP |
|---|---|---|---|
| (a) FPN | 0 | share | 75.8 |
| (b) FPN with | 0.1 | PDB | 76.3 |
| (c) FPN with | 0.3 | PDB | 76.3 |
| (d) FPN with | 0.5 | PDB | 75.8 |
| (e) FPN with | 0.7 | PDB | 76.2 |
| (f) FPN with | 0.5 | merge | 76.2 |
| (j) FPN with | ACE | merge | 75.5 |
| (h) FPN with | ACE | concat | 76.1 |
| (i) FPN with | ACE | PDB | 76.6 |

The increased detection accuracy in (b)–(e) compared with baseline (a) proves that adding context to a tight-fitting RoI is beneficial. However, different AP improvement of different context expansion ratio indicates that the expansion range has different effects on various objects. If we simply introduce a fixed context expansion ratio for all objects, it may result in mismatching for different object requirements. In the case that the detection branches are all in parallel design, the performance of our proposed ACE (i) is better than all fixed context expansion methods, which proves the effectiveness of our adaptive strategy.

We compared four detection architectures in context integration networks.

*Share* in (a) denotes the normal object detection network, which shares the fully connection layer of both classification and regression without context embedding.

*Merge* is the context embedding method proposed in [10], where the features extracted from RoI and ceRoI after RoI pooling are paired and merged by a $3 \times 3$ convolution, and then same as *Share* design. We tested a fixed context expansion ratio of 0.5, which is the same as [10] in (f). The result shows it slightly improves by 0.4 mAP. Nevertheless, we found that the accuracy is reduced when applying the adaptive context ratio to (j). One cause may be that the inconsistent expansion ratio leads to variable feature representation of ceRoI, thus it is hard to train a convolution kernel to merge the various feature pair.

*Concat* in (h) reproduces the context embedding design proposed in [20]. It sends each ceRoI into different fully-connected layers for feature generation, and then concatenates the features. The rest of the design is the same as *Share*. Since we mainly focused on the design of detection branch, for lighting parameters and equally contrasting with other methods, we concatenated one ceRoI with context expansion ratio of 0.5 with the original RoI in our experiment. Among all the architectures, *Concat* with maximal parameters only improves by 0.3 mAP. We suppose it is hard to take full advantage of this design for the integration of a few features.

Unlike the popular design of sharing parameters for classification and regression, PDB (i) offers precision improvements over traditional context embedded detection branch paradigms. Parallel branches can be trained to attentively strengthen their information integration ability for specific task. Moreover, cooperating with ACE, PDB can maximize the usage of context features, boosting detection performance by 0.8 points within the affordable calculations increase.

*3.3. Object Detection Results on KITTI*

We evaluated model performance in autonomous driving scene dataset KITTI in Table 3.

**Table 3.** Object detection results evaluated on KITTI.

| Approach | Test on | Car | Pedestrian | mAP |
|----------|---------|------|------------|------|
| (a) FPN@0.5 | KITTI | 90.3 | 78.3 | 84.3 |
| (b) EFPN@0.5 | KITTI | 90.4 | 81.0 | 85.7 |
| (c) FPN@0.75 | KITTI | 73.6 | 33.4 | 53.5 |
| (d) EFPN@0.75 | KITTI | 74.0 | 35.8 | 54.9 |

(a) and (b) show the results of FPN and EFPN, which were both tested and trained on KITTI with a 0.5 IoU threshold. The mAP of EFPN is 1.4 points higher than baseline FPN. Moreover, the AP of pedestrian, which contains more small size objects, is increased lot by 2.7 points.

The comparison of (c) and (d) shows that the performance of EFPN with an IoU threshold of 0.75 also increases by 1.4 points. We suppose that the improvement of localization accuracy is probably caused by proposed PDB. The parallel design forces each fully connected layer to attentively focus on its own task. The AP improvement of high IoU threshold denotes the increase of high-quality detections, which demonstrates the proposed EFPN is well-qualified for high-security applications such as autonomous driving. Figure 4 shows several detection examples on KITTI. We can see that more accurate boxes are generated by EFPN and more small and occluded objects have been detected.

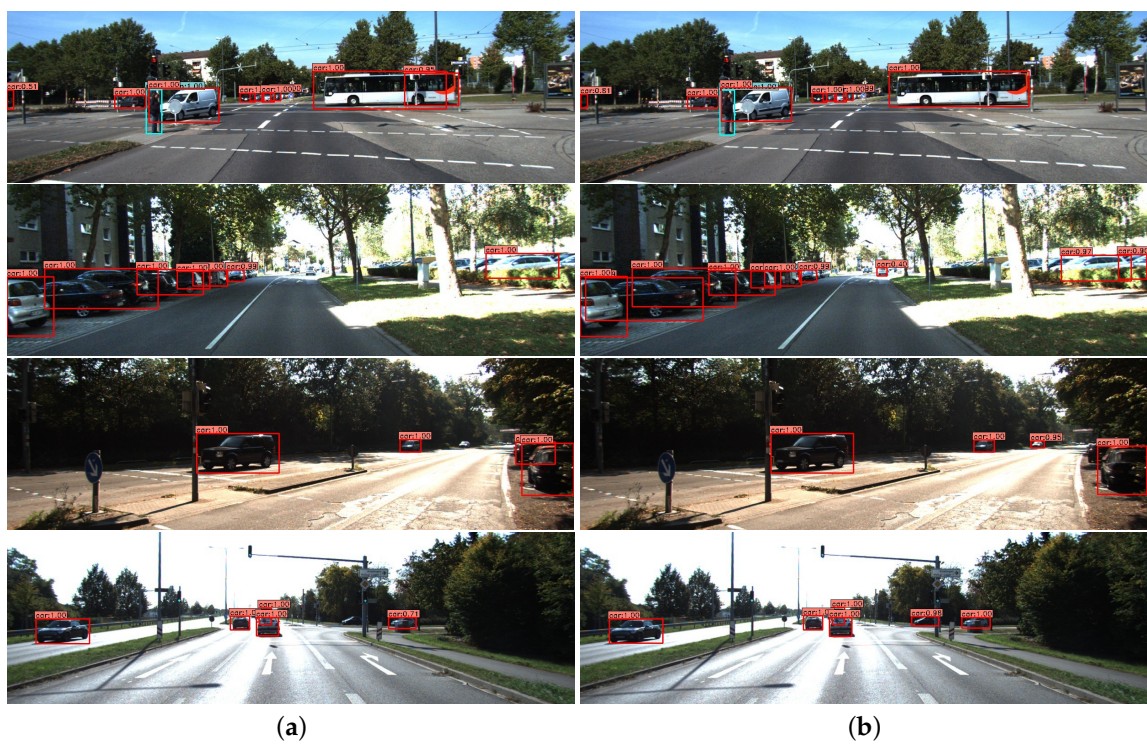

(**a**)           (**b**)

**Figure 4.** Detection examples on KITTI dataset with FPN and EFPN model. For each pair, on the left is the result of FPN (**a**) and on the right is the result of EFPN (**b**). We show detections with scores higher than 0.3.

### 3.4. Object Detection Results on Cityscapes

To verify the model generalization ability of EFPN, we trained the model on KITTI and tested on Cityscapes. The results are shown in Table 4.

**Table 4.** Object detection results evaluated on Cityscapes.

| Approach | Test on | Car | Pedestrian | mAP |
|---|---|---|---|---|
| (a) FPN@0.5 | Cityscapes | 38.5 | 19.9 | 29.2 |
| (b) EFPN@0.5 | Cityscapes | 38.9 | 21.4 | 30.1 |

EFPN is better than the baseline by 0.4, 1.5, and 0.9 points in the AP of car, pedestrian and mAP, respectively. Figure 5 shows some detection examples on Cityscapes, where more small objects and occluded objects are detected by proposed EFPN. To a certain extent, EFPN can be extended to other environments and maintain its accuracy improvement.

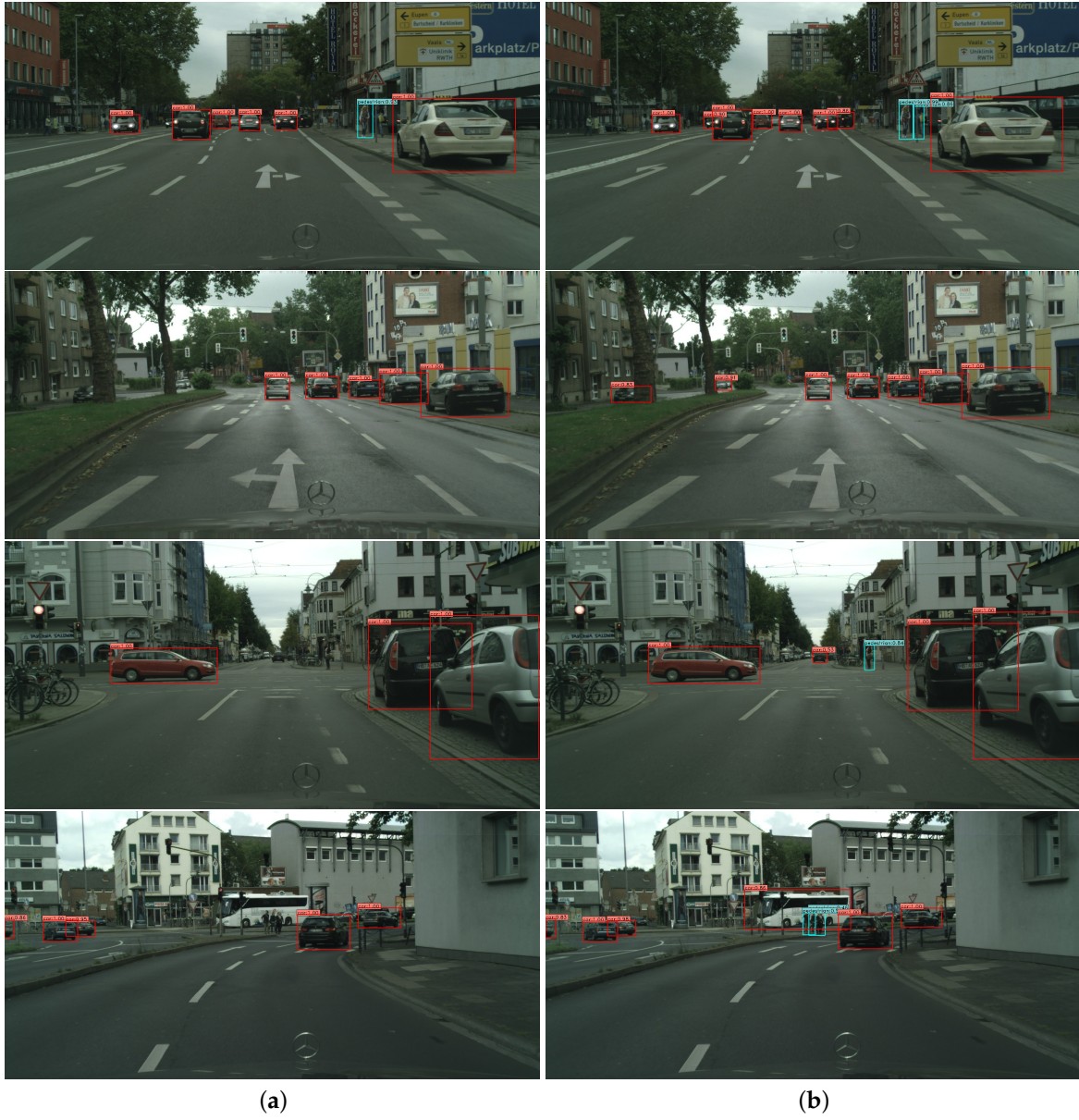

(**a**)         (**b**)

**Figure 5.** Detection examples on Cityscapes dataset with FPN and EFPN model. For each pair, one the left is the result of FPN (**a**) and on the right is the result of EFPN (**b**). We show detections with scores higher than 0.3.

Compared with the test results in KITTI, the overall APs in Cityscapes are much lower for some reasons. One cause is that the proportion of small objects and occluded objects increases greatly in Cityscapes and some of the pixel-level small object vanished after several down samplings. It may also be due to the great differences in image quality, lighting situation and traffic environment complexity between Cityscapes and KITTI.

## 4. Application in Autonomous Driving System

To further test the model practicality in real autonomous driving scenes, we installed the proposed EFPN in an autonomous driving cart for freight transportation inside a factory. The testing shows that EFPN can effectively provide surrounding object information for autonomous vehicles.

EFPN works in the perception module of autonomous driving cart. The perception module first feeds the data obtained from binocular vision camera into EFPN for object detection. Then, the results are validated by laser radar information. Next, all object information in current frame is wrapped into a ROS message and sent to decision module for further path planning and vehicle control. More broadly, EFPN supports plenty of functions such as obstacle avoidance, trajectory following and automatic parking in autonomous driving cart.

Three of the most crucial measures for object detection algorithms in autonomous driving cart are accuracy, real-time performance and vision range.

For accuracy, to satisfy production demand, the autonomous driving cart needs full-time work, including daytime, nighttime and extreme weather such as rain and snow. Besides, there are many special engineering vehicles in the factory whose appearances are significantly different from normal cars. Moreover, closely parked custom vehicles along roadside are always viewed as overlapped objects. The aforementioned challenges increase the detection difficulty and put forward higher requirements on the accuracy of object detection algorithm. The proposed deep learning model EFPN ensures the capacity of robustly detecting complex objects. For better adapting to the factory scene, we constructed a factory object detection dataset based on Pascal VOC, KITTI, and special vehicles that appear in the factory. Meanwhile, we defined a new object class *cone* for further traffic control and parking management. Overall, the factory dataset includes 21392 images, and four classes, which are *car*, *person*, *cone* and *bicycle*. We trained EFPN on the factory dataset using the same setting as KITTI.

For real-time performance, EFPN operates up to 16 fps cooperating with the 10 Hz vehicle controller, adequately ensuring the safety of autonomous driving cart.

For vision range, due to the restricted limit in the factory, the speed of the autonomous cart is lower than 10 km/h. The max braking distance is less than 0.5 m, which results in 15 m visualization demand of object detection system. Therefore, the proposed EFPN with the small object detection ability within 80 m in daytime and 50 m at nighttime well satisfies the vehicle reaction requirement within safe distances.

Figure 6 shows the real-time detection results of EFPN under some challenging scenarios. Until now, the autonomous driving cart has been operated in the factory for several months. The practical applicability of EFPN has been effectively verified.

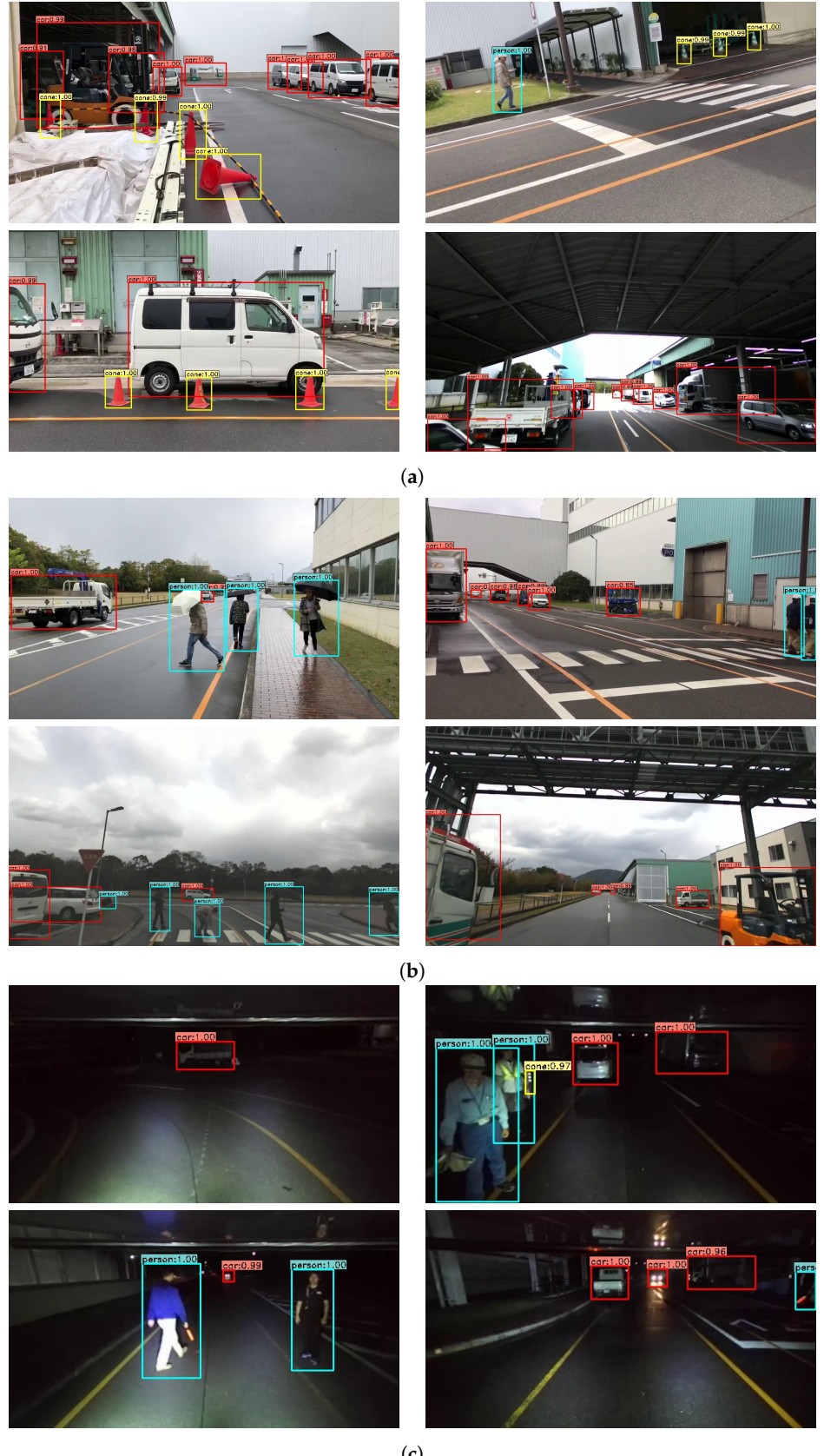

**Figure 6.** Real-time detection results in different lighting conditions and weathers in factory scenarios. (**a**) Detection results on sunny days; (**b**) Detection results on rainy days; (**c**) Detection results in the nighttime.

## 5. Conclusions

This paper proposes EFPN, a feature reinforced update based on FPN. By weighting the features in each pyramid level and adaptively utilizing the context information of object, EFPN enhances the feature expression and further improves object detection accuracy. Experiments showed that the accuracy of EFPN surpasses FPN in open datasets. The application of EFPN in autonomous driving cart proves that it can meet the demands of accuracy, efficiency and visibility in multiple scenarios.

**Author Contributions:** Conceptualization, Y.W. and S.T.; data curation, S.Z. and Y.W.; formal analysis, S.Z. and Y.W.; investigation, Y.W.; methodology, Y.W. and S.T.; software, Y.W. and S.Z.; project administration, Y.W.; resources, Y.W.; supervision, S.T. and H.O.; Validation, Y.W. and S.Z.; visualization, Y.W.; writing—original draft preparation, Y.W.; writing—review and editing, Y.W., S.T. and H.O.; and Funding acquisition, H.O.

**Funding:** This research received no external funding.

**Conflicts of Interest:** The authors declare no conflict of interest.

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
