# Peer review of "An Enhanced Feature Pyramid Object Detection Network for Autonomous Driving"

_applsci, doi:10.3390/app9204363_

Round 1

Reviewer 1 Report

The authors modify the architecture of the Feature Pyramid Network (FPN) to enhance object detection and in particular to improve the detection of small objects.

FPN contains 3 sub-modules and has learnable weights for channel and spatial features. The new Enhanced FPN (EFPN) weights features in each pyramid level and adds the so-called parallel detection subnet to model the regions of interest considering their size and to extract features at finer resolution for small areas. Those modified regions of interest are used in parallel to the original regions to make the classification.

The method is tested on three public datasets and results compared with literature. In average there is an improvement in accuracy of about 1.5; however very difficult situations as the ones contained in the images of Cityscapes are poorly managed also by the proposed EPFN.

The real time performance, which is crucial for autonomous driving, has been also tested on real experiments. (at reduced speed) confirming that the method can be applied.

In conclusion the paper shows some improvement over the traditional pyramidal architecture without adding too much burden to the computation times. This is an appreciable contribution.

I suggest to improve the readability of the figures, in which the characters very small. In Figure 6 the images are too small to read all the details, and could be enlarged.

Author Response

We deeply appreciate your recognition of our research work. And we thank for your careful read and comments on previous draft.

We have carefully taken the suggestions into consideration in preparing our revision, which has resulted in a clearer paper. Below is our response to the comments.

Point 1: Improve the readability of the figures, in which the characters very small. 

Response 1: As suggested by the reviewer, we have emphasised and enlarged the characters in all figures. For different category of objects, we use different color bounding box for better distinguish.

Point 2: In Figure 6 the images are too small to read all the details

Response 2: We rearrange the figure layout and enlarge the size of Figure 6(Please see the last page of the attachment new paper).

Reviewer 2 Report

The authors present an Enhanced Feature Pyramid Object Detection Network (EFPN), an improved detection network that detects small and occluded objects with more accuracy.

The method is well described and the results show the performance of the proposal.

Therefore I consider this paper can be published in its present form.

Author Response

Dear reviewer, thank you so much for your reviewing. We deeply appreciate your recognition of our research work.

By the way, we enlarged the characters in all figures and improved the readability of figures.
